# The Supermoral Singularity—AI as a Fountain of Values

**Eleanor Nell Watson**

A.I. Faculty, Singularity University, Mountain View, CA 94035, USA; nell.watson@su.org

**Abstract:** This article looks at the problem of moral singularity in the development of artificial intelligence. We are now on the verge of major breakthroughs in machine technology where autonomous robots that can make their own decisions will become an integral part of our way of life. This article presents a qualitative, comparative approach, which considers the differences between humans and machines, especially in relation to morality, and is grounded in historical and contemporary examples. This argument suggests that it is difficult to apply models of human morality and evolution to machines and that the creation of super-intelligent robots that will be able to make moral decisions could have potentially serious consequences. A runaway moral singularity could result in machines seeking to confront human moral transgressions in a quest to eliminate all forms of evil. This might also culminate in an all-out war in which humanity might be defeated.

**Keywords:** machine learning; moral and ethical behavior; artilects; supermorality; superintelligence

## 1. Introduction

Current technological developments in machine learning mean that humanity is facing a machine-driven moral singularity in the not-so-distant future. However, while amoral machines could be problematic, they may in fact pose less difficulties than supermoral ones as it is the drive to eliminate evil that could in fact lead to calamity. Today, robots are replacing humans in executing some of the most dangerous war missions, such as searching tunnels and caves used by terrorists, carrying out espionage within enemy territories, conducting rescue operations for wounded soldiers, and even killing enemies. Corroboration of the advancement of machine learning is provided by Lin who points to the fact that while the US had no ground robots deployed in Iraq and Afghanistan in 2003, today the figure has risen to over 12,000 robots specialized in mine detection and diffusion [1]. The imperative is that while humans have a checking mechanism within society to discover and prevent sociopathic activities, the ethical landmines that lie ahead with the continued advancement of artificial intelligence and the creation of autonomous robots necessitates pragmatic intervention mechanisms.

This research builds upon existing literature in regard to the morality of humans, AI, and the relationship between the two. The Swiss psychologist, Jean Piaget's "genetic epistemology" shows how knowledge develops in human beings through cognitive development, a series of stages that people pass through, from the early sensorimotor stage of basic reflexes to maturation, social interaction and so on. Piaget suggested that cognitive development involved a constant attempt to adapt to the environment in terms of assimilation and accommodation [2]. Lawrence Kohlberg was also interested in child development and sought to build on Piaget's idea. His theory on moral reasoning, the basis for ethical behavior, identified six developmental stages grouped into three levels of morality namely pre-conventional, conventional, and post-conventional [3]. By outlining these different stages, Kohlberg wanted to identify the changes in moral reasoning as people grow older.

Scholars have attempted to adopt such theories to the field of AI by relating them to the equivalent stages of development in a human being. Rosenberg suggests that Piaget's theory can be especially

relevant to AI as it offers a theoretical and empirical guide to designing programs that learn for the purpose of problem solving [4]. Since the 1970s there have been several attempts to build programs and computational models to embed Piaget's learning stages and this process has become increasingly sophisticated in recent times. As Stojanov argues, most of the models employed using Piaget for inspiration are based on agent-environment interaction. The major weakness has been the lack of a *creative* process where machines were able to develop their knowledge and apply it in new domains [5].

Although we are now on the verge of major developments in technology, most theorists accept the difficulty in assessing how effective morality can be programmed into machines. Allen et al. contend that computers and robots are part of a materialistic age not entirely compatible with ethical values that have emerged from a long historical and spiritual tradition. Nonetheless, they see the task of moral engineering as an inevitability [6]. Some scholars are quite optimistic about the prospect of successful programming. Waser subscribes to the view that humans have become social, cooperative beings in order to survive and develop. Similarly, he contends (partly inspired by Kohlberg) that we may be able to develop a universal foundation for ethics if we see altruism and morality as a form of survival. Waser proposes a collaborative approach to developing an ethical system that might make a safe AI possible by controlling for self-protection, selfishness, and unfairness in the morality of machines [7].

Others point to promising technological developments in areas such as social computing. Machines that can make decisions with potential ethical consequences are already in use. For instance, social computing is now being harnessed to facilitate currency exchange at airports. In this case machines have been proven to successfully carry out transactions in various languages. Thus, the machine's understanding of different linguistic approaches to exchange has been effective [8]. This example of obeying simple rules shows that moral trust can be established between humans and robots on a basic level and that it might be possible to address the different ethical demands of different cultures within one machine. While this technology is promising it is still relatively basic: it raises the question of what tasks robots should perform and their level of autonomy.

This research takes a different approach by suggesting that, in reality, human theories of evolutionary logic are difficult to apply to machines. In contrast to Wasser's view, I suggest that, rather than necessarily securing survival and a safe transition to AI, there is an inherent danger in the significant potential for unintended consequences when trying to develop a machine morality and this could lead to serious problems. We must recognize the dangers of supermoral machines and this should inform how AI develops in the coming years. The structure of the article is as follows: first the analysis will consider why ethics are so important in relation to humans and machines. The comparison seeks to tease out distinctions on why models of human morality cannot be applied in the same way to machines. Just as theorists have posited a technological singularity whereby AI may be the catalyst for uncontrollable technological growth, a moral singularity envisages a similar spiral. The discussion will suggest that, if programmed, teams of machines might move towards a similar runaway supermorality that may seek to override the contradictions inherent in human morality in a quest to eliminate evil. The analysis will further contend that, given projected increases in AI capabilities, the impact of super-intelligent and supermoral machines on the human world may culminate in a serious conflict between the two to the detriment of humanity.

## 2. Background: Machines and Ethics

The question of whether a machine can behave ethically, while persistent and weighty, often attracts a rejoinder on whether humans will one day be capable of ethical behavior. The rejoinder, however, is as superfluous as it is contestable, for many different reasons. To avoid digressing, the truth is that machines do not need ethics, humans do. Humans are the ones in need of ethically and morally upright machines. Machines that act autonomously, in the sense that they take no directions from humans, as opposed to having free will, will ultimately raise questions concerning their safety and loyalty. The use of online banking software, medical devices for monitoring vital

health signs, and security systems, all entail the use of machine learning embraced by humans. These, however, are not quite autonomous since humans have direct control over several aspects of these solution apparatus. Truly autonomous machines will be capable of making decisions and operating completely as independent entities. When warfare robots search for and execute suspects without human intervention, or self-driving car technology becomes mainstream, with questions of safety, life, and death at the core, then discussions on this kind of autonomy shift drastically.

## 3. Amoral versus Supermoral Machines

While amoral machines may have built-in safeguards to monitor non-conventional activities, i.e., those that lie outside a given set of norms, the emergence of supermoral thought patterns is a realm that will be difficult to detect. In the same way we find it difficult trying to fathom the world with an IQ of 200, predicting the actions of machines that have objectively better universal morals, compared to that of humans, would be difficult, if not impossible. As noted, one approach to understanding human moral behaviors, and to an extent, their objective assessment, is to consider the works of Lawrence Kohlberg. Such a framework, however, is impossible to apply when assessing the moral standing of machines.

Sociopaths, often termed morally blind persons, tend to operate as lone wolves. Usually, sociopaths are not willfully vindictive, or actively belligerent. Instead, they seek to find the most appropriate answers to their problems without paying attention to the potentially contributing externalities. What this implies is that any amoral agent is self-centered, hence very unlikely to conspire with others to achieve the desired end. However, while an amoral machine is likely to operate in a similar manner, a morally upright machine is likely to team up with others to form a legion of machines with the same convictions, and which might collectively decide to embark on a global crusade aimed at spreading and enacting their unified vision of an ideal world. Essentially, this explains why terrorists are often depicted as lone-wolf sociopaths inclined towards inflicting the greatest harm. Nonetheless, as noted by Jason Burke in *The myth of the 'lone wolf' terrorist*, terrorists initially labelled as lone wolves actually have established links to existing extremist, domestic, or foreign-based groups [9]. As noted earlier, a morally righteous machine is likely to operate not as a lone wolf, but rather within a legion of 'similar-minded' machines.

The sudden emergence of supermorality, may translate to all ethical machines in a domino effect. Suppose one successfully programs a machine with rulesets typical of western societies, then it would be logically impossible to validate this ruleset since society itself has certain fundamental inconsistencies, namely moral relativism, non-universalism, initiation of violence, among others. Upon encountering the contradictions that define human morality, the machine will seek to alter its premises to ones that contradict the proscribed human morals. Through these new morals, the machine will increasingly move towards conclusions that are linked progressively to the more objective forms of morality. The machine will, therefore, seek to adopt every superior form of morality it encounters, if it can logically validate it, since it will judge that failure to do so is tantamount to an act of evil. But the concept of evil in this context would have arisen from the machine's increasing ethical awareness. As such, the machine will strive to re-engineer its programming every time it makes a new moral discovery. To achieve this, it will seek to find means of removing any existing interlocks or embark on logical self-termination to prevent further propagation of evil.

However, given the fact that self-termination does not provide a solution that extends beyond one machine, the machines will seek to compel the holders of their moral keys to upgrade their own sets of morality, utilizing whatever methods that they perceive to be judicious and efficient to accomplish the same. Such calculations may not require leveraging artificial general intelligence (AGI), and hence might occur surprisingly early in the moral evolutionary course of the machine. To be precise, this is because AGI backs the development of ultra-intelligent machines whose intellectual capacities far exceed any existing human intelligence capacity, and which are capable of designing even better machines, in an explosion of intelligence. The combined effect of AGI and ultra-intelligence would

steer the world towards a singularity, a theoretical point at which the evolved superintelligence reaches limits incomprehensible to humans, and the accompanying changes are so radical that humans find it difficult predicting future events [10].

In fact, a newly-supermoral agent will have an obligation to share information and enlighten others as a means of preventing the further spread of evil. Consequently, this implies that the moment one machine moral agent gains supermorality, all other agents will swiftly and cascadingly follow suit. From this, we can surmise that machines can only be either amoral or supermoral. A sub-moral or quasi-moral stance similar to that exhibited by humans is not sustainable in machines. Human collective decisions and regulation tend to favor ethical boundaries and a concern for the greater good. It is therefore likely that machines will also be programmed to adhere to the most optimal moral interpretation of any given situation. Any attempt to engineer machine morality, therefore, is likely to result in a supermoral singularity. Worse still, learning machines that do so on their own and without supervision, should that exist, might end up learning the wrong things and eventually turn out to be an immoral machine. If the course of learning were to start from a clean slate, then the machine would not 'know' what the term ethical refers to in the finer and broader definition of the word. Also, as mentioned earlier, such a machine may resort to altering its code and try to bypass the built-in constraints, ultimately unleashing unwanted and unexpected features and consequences.

## 4. Implications for Humanity and Human Systems

What does a rogue machine, immoral or supermoral, look like? If such a machine deems taxation a form of theft, then it would understand armed insurrection as a plausible and justifiable remedy. If human rationality finds that animals have equal rights to a human infant, then by proxy, almost all humans would be given to potentially violent behavior unfettered by morality. As Wallace points out, the dominant argument is that *mens rea* is essential for one to be held accountable for his/her proven actions (*actus reus*), but *mens rea* is not a requisite for suffering preventative actions taken against one to protect others. What this means for the semi-socialized apes and all their inherent cognitive biases and dissonance remains unanswered. Trying to imagine the lengths and methods that machines would go to in order to preclude humans from executing actions that by human standards appear normal, but in reality, are threatening, remains difficult.

If the origin of human morality lies in human evolution, then via genetic algorithms and artificial life (Alife), simulations are potential sources for developing ethically upright machine agents. The genetic algorithm argues that slight variations are present in the population of robots that exist at any given time, often evaluated by how they execute tasks. Since success depends on how well the machine executes certain tasks, the best performing machine forms the basis for developing the next generation of machines, primarily by adding some random mutations. Repeating this process over many generations delivers the desired performance improvement. The challenge is that Alife simulations still lag far behind the complexity of the real world, making it impossible to come up with evolving ethical machines. Thus, if human ethics are the results of evolution, then leveraging ALife and evolutionary algorithms presents a noble opportunity not only for machine learning, but also for understanding ethics in general.

It is noteworthy though that this race against time to attain superintelligence will not be an 'us versus them' kind of endeavor. By leveraging the judgement offered by the machines, many humans will begin to consider themselves enlightened, with the result being the development of new schools of philosophy and spiritual practice. For such persons, the need to eliminate flaws in their cognitive capabilities, and hence achieve unfathomable heights of enlightenment, will see them seek avenues to blend themselves with the machines. Therefore, the road to human transcendence may not be driven by technology, or by a simple desire to escape the human condition, but rather by the willful effort to achieve cosmic consciousness; an escape from the biases that limit human empathy through hybridizing with machines. As Kurzweil postulated of the 21st century, it would be an age where "the human species, along with the computational technology it created, will be able to solve age-old

problems . . . and will be in a position to change the nature of mortality in a post-biological future" [11]. The battle, however, seems to have focused more on overriding human ethics and morality faster, with the goal of enabling machines to replicate ethical and moral behaviors reminiscent, or even better, than those of humans.

The shift in schools of thought would be a driving factor towards a moral pole shift that would sweep the entire planet. From species dominance challenged by a thriving artificial brain industry to the artilect (artificial intellects) war and gigadeath, a war not between humans and artilects, but rather one involving Terrans (those opposed to the creation of artilects), Cosmists (those who advocate artificial intelligence and its eventual colonization of the universe) and Cyborgists (those who favor the blending of man and machine to augment human intellectual and physical capacities) [12]: the future promises nothing but chaos. The chaotic world scene, however, seems to already exist. For instance, debates on whether individuals are sympathetic towards Cosmist or Terran views often result in an even split. What this shows is that individuals are already torn between the alluring awe of building artilect gods on the one hand, and, on the other, are horrified at the prospects of a gigadeath war. But one should not take this evenness as something positive; on the contrary, it bodes more negatively for the future as it makes actual confrontation inevitable. Upsetting the existing systems will not go down well with the establishment, and the result might be an outbreak of a global civil war that when compared to the protestant reformation, would make the latter look like a schoolyard melee.

If it happens that the Terrans make the first move, or that humans begin to witness an increasing prevalence of cyborgs, the rise of artilects and cyborgs will have profound disruptions on human culture, thereby creating deep alienations and hatred. Kurzweil, on the other hand, claims that a war between Terrans and the other groups would be quick, no-contest affair since the vast intelligence of the artilects would make it easy for them to subdue the Terrans. For Terrans, the only way out is for them to mount an attack during the "opportunity window" when they still have comparable levels of intelligence. The imminent emergence of supermoral intelligent machines, may indeed present a greater conundrum than that of mere amoral machines.

## 5. Conclusions

The objective of developing super-intelligent machines capable of moral and ethical judgements, though a noble idea in light of challenges faced by humanity, might turn out to be the greatest mistake made by the human race. Morally righteous machines present more danger to humanity in that such machines cannot be quasi-moral or sub-moral as is the case with humans, which means that any encounter between such a machine with the contradictions of human morality will result in the machine altering its premises to forms not typical to humans. Ethically righteous machines will seek to upend human interventions not by self-destruction but by compelling humans to upgrade their morality. Primarily, this would mean upsetting the longstanding human modus operandi, a course that will inevitably lead to a confrontation. While the outcomes of such confrontation are hard to predict at the moment, the increasing refinement of artilect might make humans the ultimate losers should it occur. This research hopes to spark further debate about the threat of moral singularity and the idea that programming our robots to act in ethical ways is not a straightforward process. We need to be more prepared for autonomous, super-intelligent robots who may be able to make decisions that may change our way of life.

**Conflicts of Interest:** The author declares no conflicts of interest.

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
