# Peer review of "The Supermoral Singularity—AI as a Fountain of Values"

_2504-2289, doi:10.3390/bdcc3020023_

Round 1

Reviewer 1 Report

[Comments]

The submitted manuscript is not a full journal article, but a viewpoint article.
It is an interesting viewpoint. The arguments have been logically put forward, and overall this is a well crafted viewpoint.  I enjoyed reading the manuscript. I would recommend the following revisions to improve the quality of the article:

1.
Author has copied lines 84-98 in the abstract of the article. The reviewer would strongly recommend the author to revise the abstract to succinctly express the gist of the viewpoint. Please avoid using different paragraphs in the abstract.

2.
Lines 115-116:
Author should briefly explain why a sub-moral or quasi-moral stance similar to humans is impossible in a machine? Reviewer feels that sub-moral stance may be possible by hard-setting certain rules which cannot be overridden. Author should add some explanation to clearly express the viewpoint for the benefit of the readers.

3.
Line 127-128:
The word 'rapists' in this context does not seem to fit.

4.
Line 143:
To avoid confusion, please mention to the abbreviation Alife on line 137 where it is first used: artificial life (Alife)

5.
Line 167:
Many readers of the article may not be aware of H. de. Garis' work. Hence, it might be appropriate to shortly define Terrans, Cosmists, and Cyborgists, so that the readers can clearly follow the contents ahead.

6.
It seems that the author has used a different font than the one used by the journal. It should be corrected.

7.
Is the word 'Sin-gularity' (with a dash) in the title deliberate, or is it a typo?

Author Response

Dear Reviewer,

Thank you very much indeed for your extremely kind and useful feedback.

I have worked to incorporate your insights into the paper revision.

Thank you again for your time, kindness, and consideration.

Sincerely,

–Nell Watson

Reviewer 2 Report

Authors of the work “The Supermoral Sin-gularity AI as a Fountain of Values”. This article presents the current problem of current technological developments in machine learning, which must take into account the moral uniqueness driven by the machine. Supermorality can produce a domino effect on all ethical machines. Faced with the contradictions that define human morality, the machine will seek to alter its premises to contradict forbidden human morality.

Overall, it is a well-structured paper; the introduction section is wide and presents the purpose of the research in detail. There is a brief background about supermoral machines. Although the proposal is interesting and within the scope of the Big Data and Cognitive Computing journal, there are different issues that should be addressed in order to improve the work.

[Major comments]

·        Improve the state of the art, Section II: Background: Machines and Ethics

[Minor comments]

·        It would be interesting to evaluate if the new systems of IoT of monitoring of context parameters and of the users like those exposed by the following works are implied by these machines of morality:

o   Sergio, A., Carvalho, S., & Marco, R. E. G. O. (2014). On the Use of Compact Approaches in Evolution Strategies. ADCAIJ: Advances in Distributed Computing and Artificial Intelligence Journal3(4), 13-23.

o   Chamoso, P., González-Briones, A., Rivas, A., De La Prieta, F., & Corchado, J. M. (2019). Social computing in currency exchange. Knowledge and Information Systems, 1-21.

·        It would be interesting to propose how these technological developments face the moral uniqueness driven and how these machines can tackle it in the future.

The used English is correct.

Author Response

(The authors gave the same response as above.)

Reviewer 3 Report

This article presents the current technological evolution in machine learning process. This means the humanity begins to face a moral singularity determined by the machine-driven in the near future. It is also described the difference between amoral machines, moral and supermoral machines and the evolution of robots that slowly will replace humans.

The article is detailed, and it provides all the information needed to understand the amoral and supermoral machines and robots evolution . The authors used and added the knowledge base needed. The article is well documented, and also, it has up-to-date references. Even if we talk about a non-technical reader or a technical one, the article is straightforward to apprehend, and at the first view, the reader will understand the main idea.

This article describes possible scenarios for the development of moral machines versus amoral machines. For super moral machines there is a problem because a machine programmed with sets of rules touting western societies is logically impossible to validate because society has inconsistent fundamentals, such as the initiation of violence.

At the same time, a moral machine tends to adopt any superior form of morality that it encounters so it judges that failure to do something is equivalent to an evil act. The article is well-structured but formatting should be according to the journal template.

Overall, the paper presents the potential to gain the reader's attention and the article is an interesting one. However, the authors should higlight their novel contributions or findings.

At the end of the "Introduction" section, there is no outline of the structure of the paper.

The authors should add future work in the conclusions.

The article needs to be revised for spelling and grammar errors.

The references should be more updated (i.e. 2018)

Author Response

(The authors gave the same response as above.)

Reviewer 4 Report

The contribution of this research work is quite new and significant and it has promising potential for the future of technological studies. 

However, the presentation of the paper should be improved.  Specifically, the major weakness of the paper is that it not presented the aim of the work ( either in the abstract or in the introduction). In addition, it is not specified the methodological approach on which the work is based.

Moreover, in order to show the contribution of this paper more clearly, the authors should provide a comparison to other analogs existing approaches published in the literature, and explain the contribution of this work compared to the state of the art. 

In particular, I think the paper could benefit by reporting some literature references for all the major concepts (e.g. singularity) mentioned in the article. Also, some major statements should be better motivated (e.g. "However, amoral machines may, in fact, be less of a problem than supermoral ones.").

Author Response

(The authors gave the same response as above.)

Round 2

Reviewer 2 Report

The article has been substantially improved with the recommended improvements. The article must use the template of the journal.

Reviewer 3 Report

All comments have been addressed

Reviewer 4 Report

The authors addressed the reviewers comments and the quality of the paper has been much improved.

I propose the article for publication.